# Lipid Metabolism and Actin Cytoskeleton Regulation Underlie Yield and Disease Resistance in Two *Coffea canephora* Breeding Populations

**DOI:** 10.3390/plants14233675

**Published:** 2025-12-03

**Authors:** Ezekiel Ahn, Sunchung Park, Jishnu Bhatt, Seunghyun Lim, Lyndel W. Meinhardt

**Affiliations:** Sustainable Perennial Crops Laboratory, Agricultural Research Service, United States Department of Agriculture, Beltsville, MD 20705, USA

**Keywords:** SNP analysis, Bootstrap Forest, leaf rust resistance, green bean yield

## Abstract

Distinct breeding populations of *Coffea canephora* often exhibit genetic divergence, yet the biological pathways underlying yield and leaf rust resistance in contrasting populations remain poorly understood. Here, we performed a comparative genomic analysis of two populations (Premature and Intermediate) to dissect the genetic architecture of coffee bean production, green bean yield, and leaf rust incidence. By integrating single-SNP association, machine learning (Bootstrap Forest), and Gene Ontology (GO) pathway analysis, we found that the Premature population’s traits were linked to specialized metabolic pathways, particularly lipid modification and organelle lumen–associated processes. In contrast, the Intermediate population was governed by core cellular machinery, with significant enrichment for actin cytoskeleton regulation and salicylic acid signaling. These findings demonstrate that distinct breeding populations achieve agronomic success through fundamentally different biological strategies and provide a reusable resource of ranked SNP lists for targeted, population-aware breeding.

## 1. Introduction

Coffee is one of the world’s most valuable agricultural commodities, supporting the livelihoods of millions globally. Production is dominated by *Coffea arabica*, known for cup quality, and *Coffea canephora* (robusta), which contributes approximately 40% of global production. *C. canephora* is increasingly vital due to its inherent disease resistance, higher yield potential, and adaptability to warmer climates [1,2,3]. However, like many perennial crops, coffee breeding faces significant challenges, primarily the long juvenile period and the need for extensive multi-year, multi-location field trials to evaluate complex traits like yield and disease resistance [4,5,6,7,8].

To accelerate genetic progress, genomic selection (GS) and genome-wide association studies (GWAS) have emerged as powerful tools to shorten breeding cycles [9,10,11,12]. While recent studies on *C. canephora* have successfully demonstrated the utility of genomic prediction [13] and polygenic association models [14], these efforts have largely focused on statistical accuracy rather than biological interpretation. Although selection history is known to drive genetic divergence between breeding populations [15,16,17], the specific biological pathways that drive agronomic performance in distinct populations remain largely unexplored.

Here, we address this gap by comparing the genetic architecture of two distinct *C. canephora* breeding populations: Premature (early ripening) and Intermediate (late ripening). To achieve this, we integrate single-SNP association, machine learning (Bootstrap Forest), and Gene Ontology (GO) enrichment to dissect the biological mechanisms underlying coffee bean production, leaf rust incidence, and green bean yield. By analyzing these populations side-by-side, we aim to determine whether they achieve agronomic success through shared or distinct biological strategies, providing targeted insights for tailored, population-specific breeding strategies.

## 2. Materials and Methods

### 2.1. Analytical Design and Rationale

To dissect the genetic architecture of complex agronomic traits, we employed a multi-layered analytical pipeline. While conventional Single-SNP association allows for the identification of specific genomic loci with major effects, it often lacks the power to capture complex, non-linear interactions inherent in polygenic traits. To overcome this limitation, we integrated a machine learning approach, Bootstrap Forest, which ranks markers based on their explanatory power (variable importance) while accounting for interactions among loci. Finally, to bridge the gap between statistical association and biological mechanism, we performed GO enrichment analysis on the top-ranked loci. This integrated approach allows us to move from identifying individual SNP markers to characterizing the distinct biological pathways driving agronomic performance in each population.

### 2.2. Experimental Populations and Data

The study analyzed two distinct *C. canephora* breeding populations, designated Premature and Intermediate, developed by the Instituto Capixaba de Pesquisa, Assistência Técnica e Extensão Rural (Incaper), ES, Brazil. These populations were selected because they represent the core heterotic groups used to extend the harvest season in the region; the Premature population ripens approximately one month earlier than the Intermediate population. The Intermediate population was derived from crosses of 16 progenitors, while the Premature population was selected from 9 progenitors [13].

The study analyzed 119 genotypes from the Intermediate population and 96 genotypes from the Premature population. In 2006, these populations (totaling 3570 and 2880 trees, respectively) were established in the field using a randomized complete block design (RCBD) with three replications. Each experimental plot consisted of five clonal plants. The trials were conducted at two locations to capture environmental variability: Marilândia Experimental Farm (FEM; 19°24′ S, 40°31′ W, 70 m altitude) and Sooretama Experimental Farm (FES; 15°47′ S, 43°18′ W, 40 m altitude). Standard agronomic practices, including fertilization and pest management, were applied uniformly across all plots and years to minimize environmental noise.

Phenotypic data were collected over four consecutive harvest years (2008–2011) for three traits: production of coffee beans (mature coffee fruit in the “cherries” stage, in 60 kg bags per hectare), yield of green beans post-harvest (ripened beans, in g, after processing), and natural infection of coffee leaf rust caused by *H. vastatrix*. Genotyping was performed using Genotyping-by-Sequencing (GBS), resulting in 45,748 SNPs for the Intermediate population and 59,332 SNPs for the Premature population after quality control. Further details on population development can be found in Ferrão et al. (2019) [13].

### 2.3. Phenotypic and Genotypic Data Acquisition

Phenotypic data were collected over four consecutive harvest years (2008–2011) [13]. Three key agronomic traits were evaluated:Production of coffee beans: Measured as the volume of mature fruit (“cherries”) harvested, expressed in 60 kg bags per hectare.Green bean yield: Measured as the weight (g) of processed, dried beans relative to the fresh harvest weight.Leaf rust incidence: Assessed visually using a 1–9 scale based on sporulation intensity, where 1 indicates absence of symptoms (resistant) and 9 indicates severe sporulation (highly susceptible). Scoring was performed during the period of high natural infection pressure to maximize discrimination between genotypes.

To ensure the validity of the maturation groups, fruit ripening was monitored throughout the study. The Premature population consistently reached harvest maturity approximately one month earlier than the Intermediate population across all four years and both locations, confirming the distinct phenotypic behavior of these groups. Standard agronomic practices, including pruning, fertilization, and pest control, were applied uniformly across all plots to minimize environmental variability.

Genotyping was performed using Genotyping-by-Sequencing (GBS). After rigorous quality control (filtering for triallelic SNPs, Minor Allele Frequency < 1%, and call rates < 70%), the final dataset comprised 45,748 SNPs for the Intermediate population and 59,332 SNPs for the Premature population.

### 2.4. Single-SNP Association Analysis

Phenotypic inputs correspond to adjusted means (BLUPs) from the mixed model described by Ferrão et al. [13], which accounts for block and year effects and filters obvious outliers. Using these adjusted means allowed us to focus the genomic analyses on the genetic signal while minimizing environmental noise. We performed a response screening analysis in JMP Pro 17 (SAS Institute Inc., Cary, NC, USA) [18], using the platform’s default settings, to identify individual SNPs associated with each trait. For each population (Premature and Intermediate) and trait combination, the adjusted phenotypic value was used as the response variable, and all SNPs were included as predictor variables. A series of individual linear regressions was performed for each SNP, and a *p*-value for the association was calculated. This approach tests the additive effect of each SNP individually, assuming a linear relationship between the number of reference alleles and the trait value. To control for the high probability of false positives inherent in testing tens of thousands of SNPs, we applied a False Discovery Rate (FDR) correction to the *p*-values. Given the low genetic differentiation previously reported between these populations (*F_ST_* = 0.0158) [13], complex population structure correction was deemed unnecessary for this initial screening, relying instead on the robustness of the adjusted phenotypic means. We chose a stringent FDR-adjusted significance threshold of 0.01 to identify a robust set of high-confidence SNP-trait associations.

### 2.5. Machine Learning for SNP Importance Analysis

#### 2.5.1. Model Rationale and Implementation

To complement the single-SNP linear regressions, which test each marker individually, we employed a machine learning approach to identify important loci by considering all SNPs simultaneously. Based on a preliminary model screening using JMP Pro 17 (SAS Institute Inc., Cary, NC, USA), the Bootstrap Forest algorithm was selected for all subsequent analyses as it consistently provided the best explanatory performance across the different traits. The Bootstrap Forest algorithm, a robust ensemble method well-suited for high-dimensional genomic data, can capture complex, non-linear, and interactive effects that may be missed by single-marker models [19,20]. Critically, our objective for using this model was not for developing a predictive tool, but for variable importance ranking, a method to identify the SNPs with the greatest explanatory power. Given that the goal was explanatory and not predictive, potential model overfitting for performance on a new dataset was not the primary concern. Instead, the model was used to generate a comprehensive and complementary set of candidate genes for our primary objective: the downstream biological pathway analysis. Regarding Linkage Disequilibrium (LD), the Random Forest algorithm is inherently robust to correlated predictors. While LD can split importance scores among correlated SNPs, our goal was to identify genomic regions of interest rather than single causal variants, making this “grouping effect” advantageous for pathway analysis.

#### 2.5.2. Model Parameters and Variable Importance

To ensure reproducibility and transparency, the Bootstrap Forest models were built using the following fixed parameters: Number of Trees = 100; Bootstrap Sample Rate = 1; Minimum Splits Per Tree = 10; Maximum Splits Per Tree = 2000; Minimum Size Split = 5; and a fixed random seed for reproducibility. The Number of Terms Sampled Per Split was adjusted based on the Number of Terms Sampled per Split to 30,333 for the Premature population and 23,086 for the Intermediate population (corresponding to approximately 50% of the total SNPs), to prioritize locus discovery over predictive accuracy. Variable importance for each SNP was then quantified using the “Portion” statistic, which represents the relative contribution of each SNP to the model. The top five SNPs with the highest “Portion” values for each trait and population were identified for further analysis.

### 2.6. Candidate Gene Identification and Gene Ontology Enrichment Analysis

To build a comprehensive list of candidate genes for pathway analysis, we leveraged the significant loci identified from our two complementary analytical frameworks: the statistically significant SNPs from the single-SNP association analysis and the top-ranked SNPs from the Bootstrap Forest importance analysis. For each significant SNP identified by the single-SNP association analysis (FDR-adjusted *p*-value < 0.01) and for the top five most important SNPs from the Bootstrap Forest analysis, we located the nearest annotated gene. This was performed using the *C. canephora* DH200-94 reference genome via Coffee Genome Hub (https://coffee-genome-hub.southgreen.fr/coffea_canephora (accessed on 21 February 2025)) [21,22]. For each identified gene, we recorded its gene ID, putative function, and the distance (in base pairs) from the associated SNP.

To investigate the broader biological pathways underlying the genetic architecture of these traits, we conducted a GO enrichment analysis. For this, a list of candidate genes was generated from the loci of the most important SNPs identified by the Bootstrap Forest models. To capture a robust biological signal for each trait-population combination, we selected the genes associated with the top 100 most important SNPs. This threshold was chosen to capture the highly predictive loci located at the upper tail of the variable importance distribution, ensuring a strong biological signal while excluding low-importance markers that represent background noise. The enrichment analysis was performed using ShinyGO v0.88 [23], which employs hierarchical clustering to reduce functional redundancy among GO terms. The analysis utilized the *Coffea canephora* gene database (AUK_PRJEB4211_v1), applying a stringent FDR cutoff of 0.05 to control for multiple testing and a pathway size filter to include GO terms containing between 2 and 5000 genes. The complete results of this enrichment analysis for all trait and population combinations are available in Appendix A.

### 2.7. Reproducibility and Software Specifications

All statistical analyses, including Single-SNP response screening and Bootstrap Forest modeling, were performed using JMP Pro 17 (SAS Institute Inc., Cary, NC, USA). To ensure reproducibility, the Bootstrap Forest models were initialized with a fixed random seed (Seed = 1). The genomic and phenotypic datasets utilized in this study are archived and accessible via Dryad (doi:10.5061/dryad.1139fm7, accessed on 21 February 2025). The *C. canephora* reference genome (DH200-94) and gene annotation database (AUK_PRJEB4211_v1) were accessed via the Coffee Genome Hub.

## 3. Results

### 3.1. Single-SNP Association Analysis of Agronomic Traits

The primary objective of this study was to compare the genetic architecture of two distinct breeding groups. Consequently, the results are presented to highlight the contrasts between the Premature and Intermediate populations across three complementary analytical layers: single-SNP associations, machine learning-based variable importance, and pathway-level biological enrichment.

This striking disparity in the number of significant associations (thousands in Premature vs. nearly none in Intermediate) implies fundamentally different genetic architectures: the Premature population exhibits an oligogenic structure with detectable major-effect loci, whereas the Intermediate population displays a highly polygenic distribution where individual SNP effects fall below the stringency threshold.

To identify SNPs associated with key agronomic traits in *C. canephora*, we performed a response screening analysis examining three traits (coffee bean production, leaf rust incidence, and green bean yield) in two populations (Premature and Intermediate). We applied an FDR-adjusted *p*-value threshold of 0.01 to identify significant SNP-trait associations.

The single-SNP analysis revealed a stark contrast in the genetic architecture between the two populations (Figure 1). No significant SNP associations were found for the production of coffee beans or the yield of green beans in the Intermediate population (FDR > 0.01). However, for leaf rust incidence in this population, a total of 23 significant SNPs were identified (17 with negative effects and 6 with positive effects) (Figure 1b). In sharp contrast, the Premature population showed thousands of significant associations across all traits. Specifically, we identified 1020 significant SNPs for the production of coffee beans (Figure 1d), 7100 SNPs for leaf rust incidence (Figure 1e), and 1850 SNPs for the yield of green beans (Figure 1f).

To visualize the genomic distribution of significant associations, we generated Manhattan plots showing the R-squared values for each significant SNP across the 11 chromosomes of *C. canephora* (Figure 2). For the production of coffee beans in the Premature population, a prominent peak was observed on chromosome 6 (Figure 2a). Significant SNPs were distributed across multiple chromosomes for leaf rust incidence in the Premature population, with particularly strong peaks on chromosomes 5 and 7 (Figure 2b). Significant SNPs were found on chromosomes 2, 5, 9, and 11 for the yield of green beans in the Premature population (Figure 2c). Leaf rust incidence in the Intermediate population exhibited significant associations on chromosomes 1, 2, 5, and 10 (Figure 2d).

Table 1 presents the candidate genes closest to the significant SNPs identified for each population-trait combination. Several of these genes have plausible connections to the traits under investigation. Among the significant SNPs for leaf rust incidence, several genes with known roles in plant defense were identified. In the Premature population, these included a putative disease resistance RPP13-like protein (*Cc04t14270.1*), an NB-ARC domain-containing protein (*Cc03t09860.1*), a peroxidase (*Cc02t30380.1*), and a chitin elicitor receptor kinase 1 (CERK1; *Cc05t03340.1*) (Table 1). RPP13-like and NB-ARC domain-containing proteins are often involved in pathogen recognition and downstream defense signaling. Peroxidases strengthen cell walls and produce reactive oxygen species during defense responses. CERK1 is a key receptor for chitin, a major component of fungal cell walls, triggering immune responses upon pathogen detection. In the Intermediate population, significant SNPs for leaf rust incidence were located near genes encoding a C2H2-type domain-containing protein (*Cc10t04730.1*), a WRKY domain-containing protein (*Cc10t04810.1*), a putative late blight resistance protein homolog R1B-16 (*Cc01t08110.1*), and a RING-type domain-containing protein (*Cc11t03510.1*) (Table 1). WRKY transcription factors are known to regulate plant defense responses [24,25], and RING-type proteins often function as E3 ubiquitin ligases, potentially involved in regulating defense signaling [26].

For the yield of green beans in the Premature population, a notable candidate gene was a putative *caffeine synthase 3* (*Cc09t06990.1*), which suggests a potential linkage between caffeine metabolism and bean characteristics that may influence overall yield. Other candidates across the traits and populations (Table 1) included involvement in diverse cellular processes (protein ubiquitination, signal transduction, and cell wall modification).

### 3.2. Bootstrap Forest Analysis of SNP Importance for Agronomic Traits

We analyzed variable importance from Bootstrap Forest models to further investigate the genetic architecture of the three agronomic traits and identify SNPs with the most significant influence on phenotype prediction. Figure 3 and Figure 4 present Manhattan plots showing the importance score (“Portion”) for each SNP across the 11 chromosomes of *C. canephora* in the Premature and Intermediate populations, respectively. For the production of coffee beans in the Premature population (Figure 3a), the most important SNPs were concentrated on chromosome 6, suggesting a region of significant influence on this trait. Among the top five candidate genes in this region were those encoding an alpha/beta-hydrolases superfamily protein (*Cc06t06270.1*) and an IPPc domain-containing protein (*Cc06t03050.1*) (Table 2). For leaf rust incidence (Figure 3b), several chromosomes exhibited SNPs with high importance scores, particularly chromosomes 5, 7, and 8. The top five candidate genes for this trait included an acyl-coenzyme A oxidase (*Cc07t15410.1*) and a hydroxyproline-rich glycoprotein family protein (*Cc08t07800.1*) (Table 2). For the yield of green beans (Figure 3c), chromosome 11 showed a prominent SNP importance, with other important SNPs distributed across several chromosomes. A gene encoding a TORTIFOLIA1-like protein 4 (*Cc11t13960.1*) was among the top five candidates for this trait (Table 2).

In the Intermediate population, the patterns of SNP importance differed from those observed in the Premature population (Figure 4). For the production of coffee beans (Figure 4a), while some SNPs on chromosomes 4 and 7 showed high importance, the overall importance scores were lower compared to the Premature population. Top candidate genes included a non-specific phospholipase C6 (*Cc04t10310.1*) and a Smr domain-containing protein (*Cc07t19900.1*) (Table 3). For leaf rust incidence (Figure 4b), chromosomes 5 and 10 exhibited prominent peaks, with a TPR_REGION domain-containing protein (*Cc05t15840.1*) and a nitrate regulatory gene2 protein (*Cc02t25100.1*) among the top candidates (Table 3). Only a few SNPs stood out for the yield of green beans, one of which was located on chromosome 4 (*Cc04t00320.1*; Conserved hypothetical protein) (Figure 4c; Table 3).

Table 2 and Table 3 list the top five candidate genes for each trait in the Premature and Intermediate populations, respectively, ranked by their importance score in the Bootstrap Forest models. It is noteworthy that some genes (*Cc06t03050.1*: IPPc domain-containing protein, *Cc05t02930.1*: TAF domain-containing protein, *Cc11t13960.1*: TORTIFOLIA1-like protein 4, *Cc02t25100.1*: Nitrate regulatory gene 2, and *Cc05t15840.1*: TPR_REGION domain-containing protein) were commonly shown as top candidates by the Bootstrap Forest models and the single-SNP analysis (see Table 1).

Collectively, the top predictors in the Premature population cluster around specific metabolic regulation (e.g., lipid and caffeine pathways), whereas the Intermediate population’s top hits are functionally diverse, involving signaling and structural components, consistent with a broader polygenic base.

### 3.3. Comparative Analysis and Identification of Consensus Candidate Genes

To further elucidate the biological functions underlying the distinct genetic architectures of the Premature and Intermediate populations, a GO enrichment analysis was conducted. The analysis focused on genes associated with the top 100 predictive SNPs identified by the Bootstrap Forest models for each population, assessing both individual and combined effects.

The results revealed highly distinct biological signatures for each population. In the Premature population, yield-related traits were significantly enriched for specialized cellular functions (Figure 5a,b). The analysis combining all traits highlighted ‘lipid modification’ (GO:0030258) and pathways related to the internal space of organelles, such as ‘membrane-enclosed lumen’ (GO:0031974) and ‘organelle lumen’ (GO:0043233) (Figure 5c).

In contrast, the analysis of the Intermediate population showed a strong and consistent enrichment for pathways involved in regulating the actin cytoskeleton across all trait combinations. Key enriched terms included ‘reg. of actin filament depolymerization’ (GO:0030834), ‘negative regulation of actin filament polymerization’ (GO:0030837), and ‘actin filament capping’ (GO:0051693) (Figure 6). Furthermore, combining yield traits introduced significant enrichment for ‘response to salicylic acid’ (GO:0009751) (Figure 6b), and the further addition of rust resistance highlighted a broader ‘regulation of defense response’ (GO:0031347) pathway (Figure 6c)

To identify overarching biological themes shared between the two populations, a combined GO analysis was performed by pooling the gene lists. Notably, while neither population showed significant GO term enrichment for rust resistance individually, the combined analysis revealed a strong and statistically significant enrichment for ‘chitinase activity’(GO:0004568), ‘chitin metabolic proc.’ (GO:0006030 & 0006032), and ‘programmed cell death’ (GO:0034050) (Figure 7a). This indicates a shared, high-level defense function that becomes apparent only when data from both populations are pooled. For yield-related traits, the combined analysis was predominantly characterized by pathways involved in actin filament regulation (Figure 7b,c), demonstrating that this strong theme from the Intermediate population defines the combined genetic signal. Interestingly, when all traits from both populations were combined, the enrichment profile shifted to highlight different pathways involved in phosphatidylinositol metabolism and sulfate transmembrane transport (Figure 7d). In summary, these contrasting enrichment profiles demonstrate that while the Premature population relies on specific metabolic adaptations (lipid/lumen), the Intermediate population achieves similar agronomic outcomes through the modulation of fundamental cellular mechanics (actin/signaling).

## 4. Discussion

Our study provides a new layer of biological interpretation that complements and expands upon previous work that established the potential for genomic prediction [13] and dissected trait stability [27] in these important *C. canephora* populations. Whereas prior analyses focused on the statistical accuracy of predictive models or the predictability of stability metrics across environments, our primary objective was to dissect the functional basis of the underlying genetic architecture. By integrating single-SNP regression, machine learning for variable importance, and a novel comparative GO framework, our work moves beyond statistical observation to mechanistic insight. The central finding of this study, that the Premature and Intermediate populations employ fundamentally different biological strategies to achieve agronomic performance, provides a new biological context for breeding efforts that was previously absent from the literature on these populations. These contrasts indicate that population-aware interpretation is essential: the same trait class can be underpinned by specialized metabolism in one breeding population and by core cellular machinery in another. The dramatic difference in SNP discovery rates suggests that the Premature population is governed by an oligogenic architecture, likely driven by strong selection pressure on its specialized metabolic traits (e.g., lipid modification) [15,16]. In contrast, the Intermediate population follows an infinitesimal model (polygenic), where phenotypic variance is controlled by many small-effect loci that collectively drive the “Core Cellular Machinery” but individually lack the statistical power to be detected by single-SNP methods. This explains why the Machine Learning approach (which captures cumulative small effects) was essential for characterizing the Intermediate population. This distinction dictates tailored breeding strategies: the Premature population, with its oligogenic architecture, is a prime candidate for Marker-Assisted Selection (MAS) targeting specific high-effect loci. Conversely, the Intermediate population, governed by a polygenic background, would benefit most from GS models capable of capturing genome-wide small effects that escape traditional single-marker detection [28,29].

Our analysis confirms that the Premature and Intermediate populations possess distinct genomic architectures. This difference is likely attributable to their unique selection histories, which can alter allele frequencies and narrow the genetic base over time [15,16,17]. Additionally, potential differences in linkage disequilibrium patterns [30,31,32], which have been empirically shown to differ between populations based on breeding history [31,32,33,34], and founder effects from their establishment, which are known to shape the genetic makeup of breeding populations [35], may persist and contribute to these population-specific genetic effects [36]. While our single-SNP analysis highlighted a greater number of significant associations in the Premature population, our new pathway-level analysis provides a deeper biological explanation for these differences, moving beyond statistical observation to mechanistic interpretation.

The GO analysis revealed that the two populations employ fundamentally different biological strategies to achieve their agronomic traits. In the Premature population, yield-related traits were significantly enriched for specialized cellular pathways, including ‘lipid modification’ and pathways related to the ‘membrane-enclosed lumen’ of organelles. This is biologically significant, as the vacuole is the primary site for sugar and metabolite storage, and its proper function is crucial for determining overall plant yield [37,38]. Indeed, direct manipulation of vacuolar transporters has been shown to increase seed yield in model plants [39]. This pathway-level finding strongly supports our identification of a putative *caffeine synthase 3* as a key candidate gene for green bean yield, as the biosynthesis and accumulation of purine alkaloids like caffeine is a key feature of seed and fruit development in *Coffea* [40,41,42,43]. In stark contrast, the genetic architecture of the Intermediate population was defined by the GO enrichment of pathways related to ‘actin cytoskeleton regulation’. The actin cytoskeleton is fundamental for coordinating all aspects of plant growth, including cell expansion, intracellular transport, and the maintenance of structural integrity [44]; therefore, its enrichment suggests that genetic variation in the Intermediate population influences yield through the overall efficiency of core cellular machinery. This aligns with our identification of candidate genes such as *NPC6*, a non-specific phospholipase C known to be integral to lipid metabolism and seed oil production [45,46,47], and TPR_REGION domain-containing proteins, which function as core scaffolds for protein–protein interactions within essential cellular complexes involved in protein transport and hormone-mediated stress responses [48,49,50,51,52]. Taken together, the enrichment of organelle lumen and lipid modification pathways in the Premature population and actin-cytoskeleton regulation in the Intermediate population point to distinct routes that may converge on sink strength and carbohydrate partitioning.

A similar divergence in strategy was observed for leaf rust resistance. Our GO analysis of the Intermediate population highlighted the ‘salicylic acid signaling’ pathway, linking its genetic architecture to a known hormonal defense response network that is critical for systemic acquired resistance against biotrophic pathogens [53,54,55,56]. This provides a biological context for candidate genes like the nitrate regulatory gene 2, suggesting a connection between nutrient status and defense signaling in this population [57]. In contrast, the Premature population’s resistance was associated with classical defense genes like RPP13-like protein [58,59,60,61], the NB-ARC domain, which functions as an essential signaling switch in NLR immune receptors [62,63,64,65], and CERK1, a pattern-recognition receptor that is essential for perceiving fungal chitin to trigger PAMP-triggered immunity [66,67,68,69]. The power of a comparative approach was most evident when data from both populations were pooled; only then did a shared, overarching ‘defense response’ theme become statistically significant. This demonstrates that while the specific genetic components differ, they contribute to a common biological function for rust resistance across a broader genetic background. Furthermore, the final combined analysis highlighted pathways of ‘phosphatidylinositol metabolism’ and ‘sulfate transport’ (Figure 7d), suggesting that fundamental membrane signaling and nutrient transport may represent a core biological system underpinning the general fitness and performance across all measured traits. Notably, a chitin/chitinase–programmed cell death module only reached significance when gene lists were pooled across populations, suggesting a convergent downstream immunity axis that single-population analyses may miss. This helps reconcile the Premature population’s CERK1/NLR emphasis with the Intermediate population’s salicylic acid–signaling enrichment.

Our results both complement and expand upon the recent polygenic GWAS analysis by Ferrão et al. [14]. While their Bayesian Sparse Linear Mixed Model (BSLMM) identified major QTL for traits like leaf blight and plant architecture, our use of complementary methods and a population-specific focus provides deeper biological context. Where our findings converge on similar genomic regions, our pathway-level analysis offers a mechanistic hypothesis for why these regions are important, a strategy that has proven effective for revealing molecular mechanisms when integrating GWAS with other functional data [69,70,71]. The novel identification of distinct, pathway-level strategies (specialized metabolism vs. core cellular machinery) is a key contribution that builds upon these previous genomic studies by explaining the biological nature of the genetic variation (Figure 8). Our work also aligns with other studies confirming the complex, polygenic nature of traits like rust resistance in perennial crops such as *C. canephora* [72,73,74,75,76,77,78].

This study is not without limitations. The machine learning models were developed without an independent validation dataset due to sample size constraints, meaning the results demonstrate explanatory power rather than confirmed predictive ability. This is a critical consideration, as robust cross-validation is essential for accurately assessing and comparing the performance of genomic models [79,80,81]. Furthermore, we can only demonstrate statistical associations between SNPs and traits, and not definitively prove causation, a common challenge in moving from GWAS loci to causal genes [82,83]. The candidate genes we’ve identified are promising targets, but functional validation is essential to confirm their roles, a critical next step for all discovery-based genomic studies [78,83,84,85,86].

The findings from our GO analysis, however, provide clear, targeted avenues for future work. Functional validation should now prioritize not only single candidate genes like *caffeine synthase 3* in the Premature population, but also key regulators within the actin filament polymerization and salicylic acid signaling pathways now identified in the Intermediate population [84]. A deeper investigation of genotype-by-environment interactions, a well-established challenge in coffee breeding, is also crucial [9,13]. By integrating complementary analytical approaches with pathway-level analysis, this work provides a more nuanced, population-specific understanding of the genetic basis of key coffee traits and paves the way for more efficient and targeted coffee breeding programs.

## 5. Conclusions

This study demonstrates that *C. canephora* breeding populations can diverge not only in allele frequencies but in the fundamental biological strategies driving agronomic performance. By integrating multi-layered genomic analyses, we identified that the Premature population leverages specialized metabolic pathways (lipid/organelle lumen) via an oligogenic architecture, whereas the Intermediate population relies on core cellular machinery (actin/signaling) through a polygenic framework. These findings challenge the “one-size-fits-all” approach to molecular breeding, suggesting that breeding strategies must be population-specific: Marker-Assisted Selection for the metabolically driven Premature group and GS for the polygenic Intermediate group. A key limitation of this study is the reliance on data from specific environments; future work must validate these pathway-level mechanisms across diverse agro-ecological zones to confirm their stability and broader applicability.

## Figures and Tables

**Figure 1 plants-14-03675-f001:**
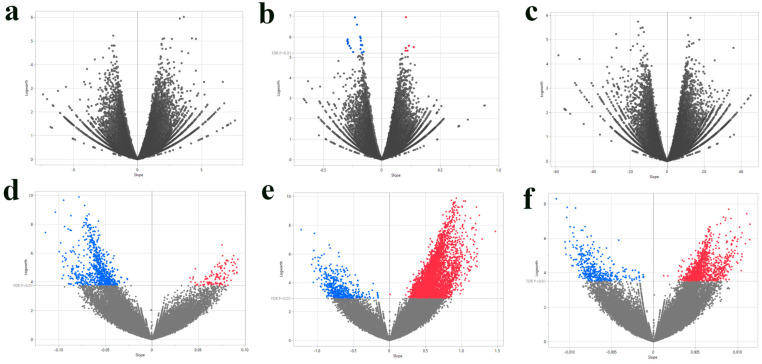
Volcano plots of single-SNP association analysis for three agronomic traits in two *C. canephora* populations. Each point represents a single SNP. The *x*-axis shows the estimated slope (effect size) from a linear regression of the phenotype on the SNP genotype, and the *y*-axis shows the negative base-10 logarithm of the *p*-value from the association test. SNPs are colored based on the direction of their effect and statistical significance after applying an FDR correction to account for multiple testing. Red points indicate SNPs with a positive effect and FDR-adjusted *p*-value < 0.01; blue points indicate SNPs with a negative effect and FDR-adjusted *p*-value < 0.01; gray points do not meet the significance threshold. The plots highlight a substantially larger number of significant associations in the Premature population compared to the Intermediate population. (**a**) Production of coffee beans, Intermediate. (**b**) Leaf rust incidence, Intermediate. (**c**) Yield of green beans, Intermediate. (**d**) Production of coffee beans, Premature. (**e**) Leaf rust incidence, Premature. (**f**) Yield of green beans, Premature.

**Figure 2 plants-14-03675-f002:**
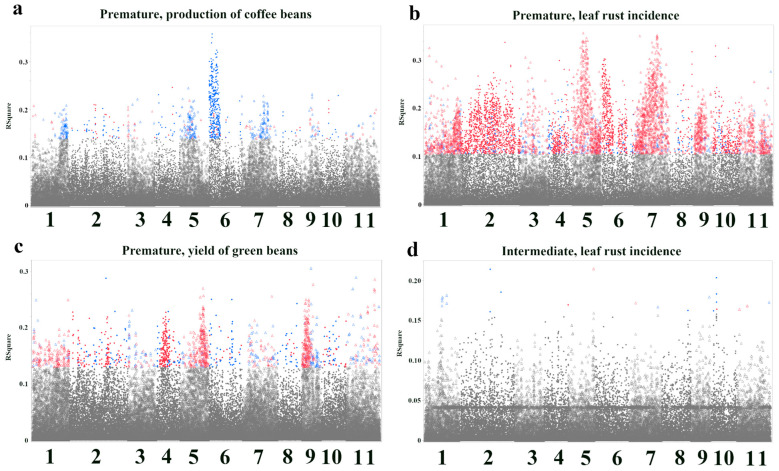
Manhattan plots of single-SNP association analysis, showing only SNPs that reached genome-wide significance (FDR < 0.01). Each point represents a single SNP. The *x*-axis shows the chromosomal location (1 through 11). The *y*-axis shows the R-squared value, representing the proportion of phenotypic variance explained by each individual SNP. SNPs are colored based on the direction of their effect: red indicates a positive effect, and blue indicates a negative effect. Gray points represent SNPs that did not reach the significance threshold. Each panel reveals the genomic regions most strongly associated with the respective trait. (**a**) Production of coffee beans, Premature. (**b**) Leaf rust incidence, Premature. (**c**) Yield of green beans, Premature. (**d**) Leaf rust incidence, Intermediate. Plots for two trait-population combinations are not shown due to a lack of significant associations.

**Figure 3 plants-14-03675-f003:**
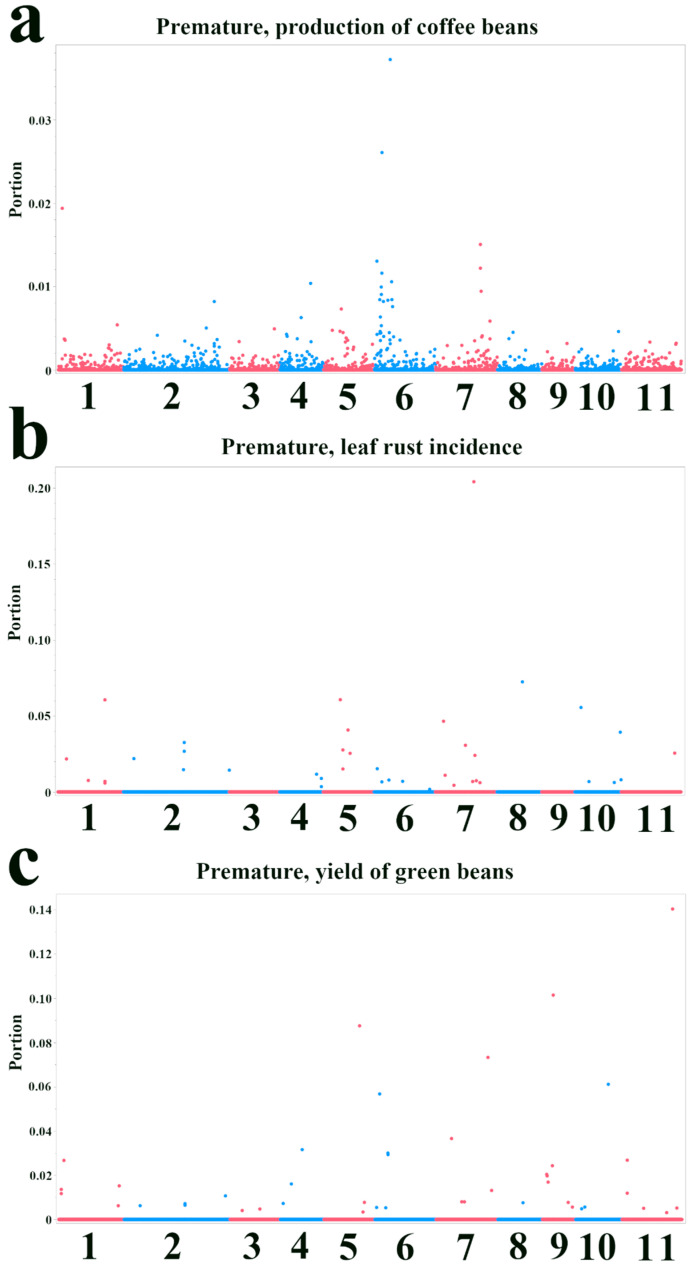
Manhattan plots showing variable importance from Bootstrap Forest models for agronomic traits in the Premature population. The *x*-axis indicates the chromosomal position of each SNP. The *y*-axis represents the variable importance score (“Portion”), which quantifies the relative contribution of each SNP to the model’s explanatory power. Higher values indicate greater importance. Points are colored in alternating red and blue to distinguish between adjacent chromosomes. (**a**) Production of coffee beans. (**b**) Leaf rust incidence. (**c**) Yield of green beans.

**Figure 4 plants-14-03675-f004:**
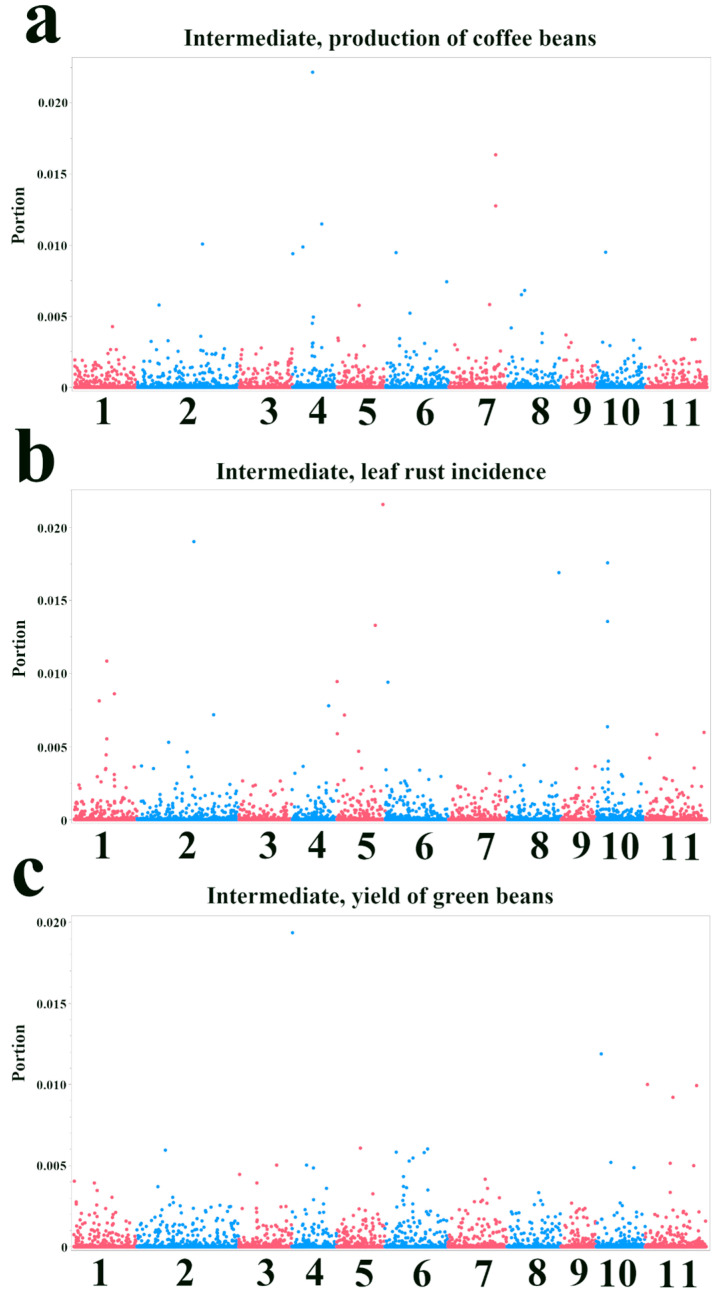
Manhattan plots showing variable importance from Bootstrap Forest models for agronomic traits in the Intermediate population. The *x*-axis indicates the chromosomal position of each SNP. The *y*-axis represents the variable importance score (“Portion”), which quantifies the relative contribution of each SNP to the model’s explanatory power. Higher values indicate greater importance. Points are colored in alternating red and blue to distinguish between adjacent chromosomes. (**a**) Production of coffee beans. (**b**) Leaf rust incidence. (**c**) Yield of green beans.

**Figure 5 plants-14-03675-f005:**
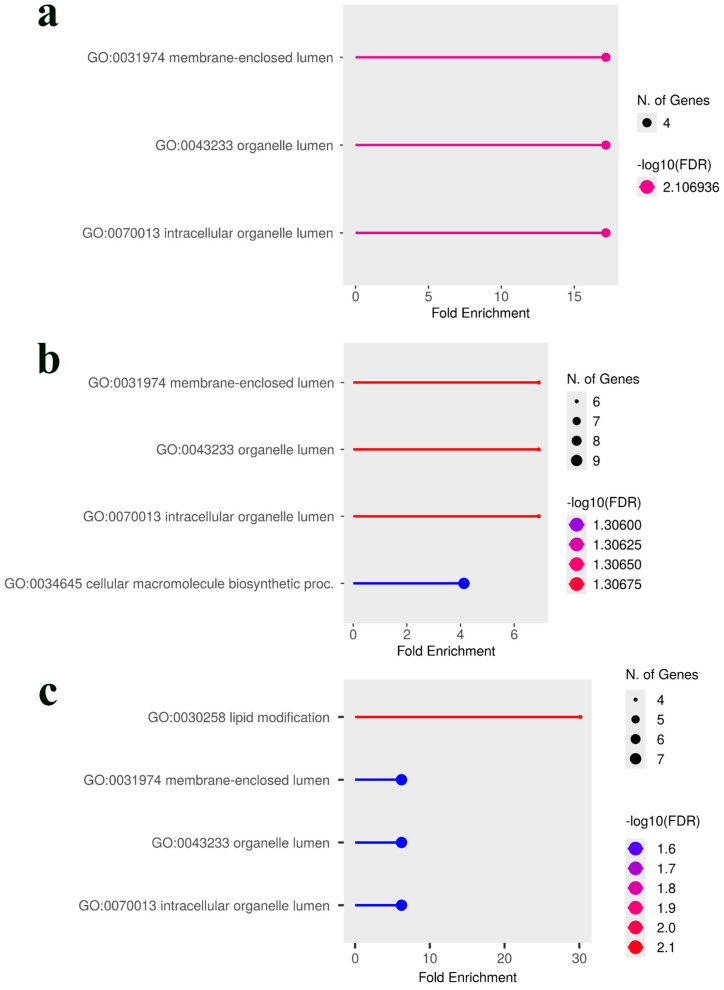
GO enrichment analysis of candidate genes in the *C. canephora* Premature population. The analysis was performed using genes associated with the top 100 predictive SNPs from the Bootstrap Forest models for different trait combinations. The plots illustrate the most significantly enriched biological pathways. (**a**) Enriched GO terms for genes associated with green bean yield, highlighting pathways related to the internal space of organelles. (**b**) Enriched terms for the combined traits of coffee bean production and green bean yield, which include the addition of a ‘cellular macromolecule biosynthetic process’. (**c**) Enriched terms for the combination of all three traits (coffee bean production, green bean yield, and leaf rust incidence), identifying ‘lipid modification’ as another key pathway. In all plots, the *x*-axis represents the fold enrichment of the term, the size of each point corresponds to the number of genes associated with the term, and the color indicates the statistical significance based on the −log10(FDR). Abbreviations used: reg., regulation; proc., process.

**Figure 6 plants-14-03675-f006:**
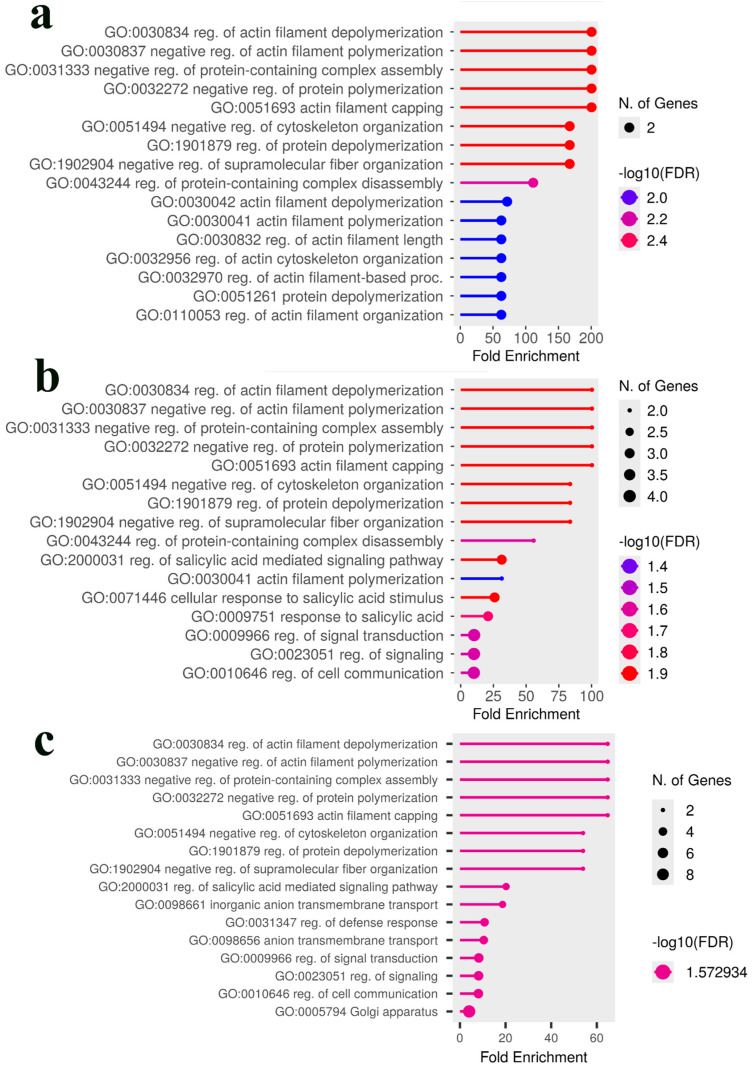
GO enrichment analysis of candidate genes in the *C. canephora* Intermediate population. The analysis was performed on genes associated with the top 100 predictive SNPs from the Bootstrap Forest models for various trait combinations. (**a**) For the green bean yield trait, the analysis reveals a strong enrichment for pathways involved in the regulation of the actin cytoskeleton. (**b**) Combining coffee bean production and green bean yield traits retains the actin-related terms and introduces significant enrichment for signaling pathways, including ‘response to salicylic acid’. (**c**) The analysis of all three traits combined (coffee bean production, green bean yield, and leaf rust incidence) reinforces the importance of defense-related signaling (‘salicylic acid mediated signaling pathway’, ‘regulation of defense response’) alongside the persistently significant actin regulation pathways. In all plots, the *x*-axis represents the fold enrichment, the point size corresponds to the number of genes, and the color indicates statistical significance based on the −log10(FDR). Abbreviations used: reg., regulation; proc., process.

**Figure 7 plants-14-03675-f007:**
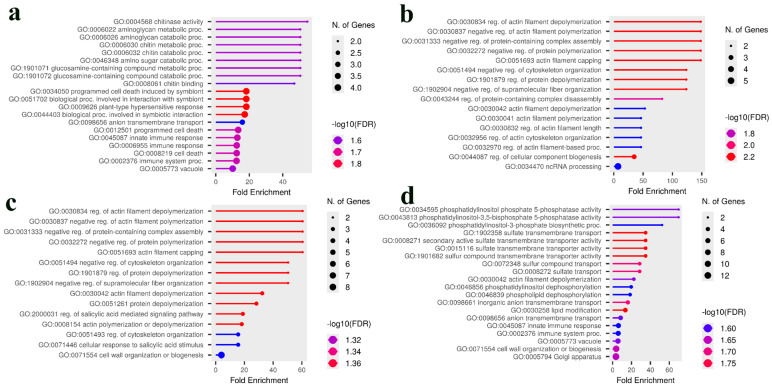
GO enrichment analysis of candidate genes from the combined Premature and Intermediate populations. The analysis was performed on pooled gene lists associated with up to the top 100 predictive SNPs from the Bootstrap Forest models in both populations. (**a**) The combined analysis for leaf rust resistance reveals a strong enrichment for defense-related pathways, including ‘chitinase activity’ and ‘immune response’. (**b**) For the combined green bean yield trait, the analysis is characterized by a strong enrichment for pathways related to ‘actin cytoskeleton regulation’. (**c**) The addition of coffee bean production traits retains the dominant actin regulation theme while also introducing pathways related to ‘salicylic acid-mediated signaling’. (**d**) When all three traits (yield, green bean, and rust) are combined, the enrichment profile highlights pathways involved in ‘phosphatidylinositol metabolism’ and ‘sulfate transmembrane transport’. In all plots, the *x*-axis represents the fold enrichment, the point size corresponds to the number of genes, and the color indicates statistical significance based on the −log10(FDR). Abbreviations used: reg., regulation; proc., process.

**Figure 8 plants-14-03675-f008:**
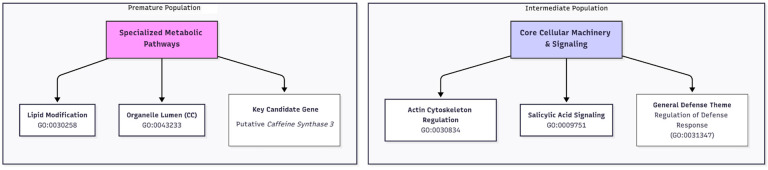
A conceptual model summarizing the distinct biological strategies for agronomic performance in the Premature and Intermediate *Coffea canephora* populations. The model visually contrasts the key findings from the comparative genomic analysis. The genetic architecture of the Premature population is linked to specialized metabolic pathways, including “lipid modification” and the cellular component “organelle lumen,” and is highlighted by a key candidate gene, a putative *caffeine synthase 3*. In contrast, the Intermediate population’s traits are governed by variation in core cellular machinery & signaling, with significant enrichment for pathways like “actin cytoskeleton regulation” and “salicylic acid signaling.” This figure illustrates that the two populations achieve agronomic success through fundamentally different, population-specific biological routes.

**Table 1 plants-14-03675-t001:** Candidate genes associated with SNPs showing significant associations with agronomic traits in two *C. canephora* populations (Premature and Intermediate).

SNP ID	Nearest Gene and Function	Base Pairs Away	FDR *p*-Value	Effect Size	R-Square
**Premature, production of coffee beans—Response Screening**					
6.2295851	*Cc06t02920.1* ADF-H domain-containing protein	0	2.12 × 10^−10^	0.49	0.35
6.2378930	*Cc06t03050.1* IPPc domain-containing protein	0	1.27 × 10^−10^	0.49	0.36
6.2075614	*Cc06t02620.1* E3 ubiquitin-protein ligase	0	0.00000000049	0.48	0.34
6.4097514	*Cc06t05200.1* NPL domain-containing protein	−29	0.0000000025	0.46	0.32
6.4477759	*Cc06t05570.1* IU_nuc	0	0.0000000026	0.47	0.32
4.15969587	*Cc04t13140.1* UDP-glycosyltransferase 83A1	−25,832	0.00000027	0.41	0.25
10.6890851	*Cc10t07860.1* GH10 domain-containing protein	0	0.0000015	0.39	0.22
5.13748592	*Cc05t03040.1* Putative Short-chain dehydrogenase reductase ATA1	−47,435	0.0000015	0.39	0.22
2.17352904 2.17352905	*Cc02t19180.1* Stress enhanced protein 1, chloroplastic	0	0.0000026	0.38	0.21
2.18145802	*Cc02t20300.1* SCP domain-containing protein	+3008	0.0000027	0.38	0.21
**Premature, leaf rust incidence—Response Screening**					
7.15362801	*Cc07t18070.1* Hexosyltransferase	−1304	0.000000021	0.54	0.29
11.32751026	*Cc11t16690.1* Urease	0	0.000000038	0.53	0.28
4.20658374	*Cc04t14270.1* Putative disease resistance *RPP13-like* protein 3	−32,177	0.00000023	0.5	0.25
3.12306478	*Cc03t09860.1* * NB-ARC * domain-containing protein	0	0.00000039	0.49	0.24
7.13312636	*Cc07t16380.1* Conserved hypothetical protein	+1562	0.00000058	0.49	0.23
2.35607328	*Cc02t30380.1* Peroxidase	0	0.0000014	0.59	0.34
5.13342765	*Cc05t02930.1* TAF domain-containing protein	+27,160	0.0000014	0.59	0.34
5.14130887	*Cc05t03220.1* Lycopene beta/epsilon cyclase protein	−3241	0.0000014	0.59	0.34
5.14494464 5.14494471 5.14494484	*Cc05t03270.1* AT1G05060.1	0	0.0000014	0.58	0.34
5.14625364	*Cc05t03340.1* Chitin elicitor receptor kinase 1	0	0.0000014	0.58	0.34
**Premature, yield of green beans—Response Screening**					
11.12510898	*Cc11t03410.1* Protein of unknown function (DUF789)	+10,726	0.000000016	0.52	0.29
2.24864284	*Cc02t27160.1* Vicianin hydrolase	0	0.000000018	0.52	0.29
9.6114101	*Cc09t05750.1* T-complex protein 1 subunit gamma	0	0.0000000052	0.54	0.31
11.7875427	*Cc11t02410.1* RING-type domain-containing protein	0	0.000000063	0.5	0.27
1.3497181	*Cc01t01920.1* SKP1-like protein 4	+5743	0.00000023	0.48	0.25
11.30714537	*Cc11t13980.1* HDAC_interact domain-containing protein	0	0.000000021	0.52	0.29
9.8293367	*Cc09t06990.1* Putative caffeine synthase 3	0	0.000000038	0.51	0.28
11.30518300	*Cc11t13720.1* GSDH domain-containing protein	0	0.000000067	0.5	0.27
11.30697733	*Cc11t13960.1* TORTIFOLIA1-like protein 4	0	0.000000071	0.5	0.27
5.26247026	*Cc05t12380.1* Transducin/WD40 repeat-like superfamily protein	−3088	0.00000006	0.5	0.27
**Intermediate, leaf rust incidence—Response Screening**					
2.22416916	*Cc02t25100.1* Nitrate regulatory gene 2 protein	0	0.00000011	0.56	0.21
10.3678570	*Cc10t04730.1* C2H2-type domain-containing protein	0	0.00000026	0.54	0.2
10.3747377	*Cc10t04810.1* WRKY domain-containing protein	0	0.0000012	0.52	0.18
1.26450374 1.26450396	*Cc01t08110.1* Putative late blight resistance protein homolog R1B-16	0	0.0000016	0.51	0.18
1.29976260 1.29976261 1.29976262	*Cc01t11280.1* Conserved hypothetical protein	−2741	0.0000013	0.51	0.18
5.28610373	*Cc05t15840.1* TPR_REGION domain-containing protein	0	0.00000011	0.56	0.21
7.3420157	*Cc07t04860.1* AAA domain-containing protein	0	0.0000027	0.50	0.17
11.13698759 11.13698762	*Cc11t03510.1* RING-type domain-containing protein	−337,600	0.0000035	0.50	0.17
4.27729192	*Cc04t17060.1* BHLH domain-containing protein	−85	0.0000032	0.50	0.17
10.3876532	*Cc10t04930.1* SASA domain-containing protein	−2641	0.0000048	0.49	0.16

SNPs were identified as significant based on a response screening analysis in JMP Pro 17, with an FDR-adjusted *p*-value threshold of 0.01. For each significant SNP, the table lists: the SNP identifier (SNP ID) (blue = negative and red = positive association); the nearest gene and its putative function (based on the annotation of the *C. canephora* reference genome); the distance in base pairs between the SNP and the start of the nearest gene (0 indicates the SNP is within the gene and negative and positive distances denote SNPs located upstream and downstream of the gene’s transcription start site (TSS), respectively); the FDR-adjusted *p*-value; the estimated effect size (slope from a linear regression of the phenotype on the SNP genotype); and the R-squared value (proportion of phenotypic variance explained by the SNP). Genes are listed separately for each population and trait combination. Only traits with at least one significant SNP are included.

**Table 2 plants-14-03675-t002:** Top five candidate genes identified by Bootstrap Forest models for three agronomic traits in the Premature population.

SNP ID	Nearest Gene and Function	Base Pairs Away	Importance Score (Portion)
**Premature, production of coffee beans—Bootstrap Forest**			
6.4939167	*Cc06t06270.1* Alpha/beta-Hydrolases superfamily protein	0	0.037
6.2378930	*Cc06t03050.1* IPPc domain-containing protein	+172	0.026
1.2020393	*Cc01t01290.1* Putative 60S ribosomal protein L23a-1	+6605	0.019
7.15086633	*Cc07t17840.1* *NB-ARC* domain-containing protein	+19	0.015
6.1079450	*Cc06t01300.1* HEAT repeat-containing protein	0	0.013
**Premature, leaf rust incidence—Bootstrap Forest**			
7.12204503	*Cc07t15410.1* Acyl-coenzyme A oxidase	+308	0.2
8.21181446	*Cc08t07800.1* Hydroxyproline-rich glycoprotein family protein	0	0.07
5.13342765	*Cc05t02930.1* TAF domain-containing protein	+27,160	0.06
1.32477500	*Cc01t14390.1* Increased DNA methylation like	0	0.06
10.1733570	*Cc10t02280.1* OBG-type G domain-containing protein	0	0.056
**Premature, yield of green beans- Bootstrap Forest**			
11.30697733	*Cc11t13960.1* TORTIFOLIA1-like protein 4	0	0.14
9.3350785	*Cc09t03950.1* NAD(P)-binding Rossmann-fold superfamily protein	0	0.1
5.24257788	*Cc05t09820.1* Glucose-1-phosphate adenylyltransferase	−1708	0.088
7.20811138	*Cc07t20130.1* Protein of unknown function (DUF1365)	+26,561	0.073
10.20833232	*Cc10t11950.1* tRNA (guanine(37)-N1)-methyltransferase	0	0.061

SNPs were ranked based on their variable importance (“Portion”) in the models, with the corresponding genomic locations shown in Figure 3. The table lists the SNP identifier (SNP ID); the nearest gene and its putative function; the distance in base pairs between the SNP and the gene’s start site (0 indicates the SNP is within the gene and negative and positive distances correspond to upstream and downstream positions relative to the TSS); and the importance score from the model.

**Table 3 plants-14-03675-t003:** Top five candidate genes identified by Bootstrap Forest models for three agronomic traits in the Intermediate population.

SNP ID	Nearest Gene and Function	Base Pairs Away	Importance Score (Portion)
**Intermediate, production of coffee beans—Bootstrap Forest**			
4.9597530	*Cc04t10310.1* Non-specific phospholipase C6	0	0.022
7.20000085 7.20000113	*Cc07t19900.1* Smr domain-containing protein	−737	0.016
4.15068413	*Cc04t12970.1* Alpha-N-acetylglucosaminidase	+804	0.012
2.26624120	*Cc02t28190.1* BHLH domain-containing protein	+1083	0.01
4.3864735	*Cc04t05180.1* Phytocyanin domain-containing protein	0	0.01
**Intermediate, leaf rust incidence—Bootstrap Forest**			
5.28610373	*Cc05t15840.1* TPR_REGION domain-containing protein	0	0.022
2.22416916	*Cc02t25100.1* Nitrate regulatory gene2 protein	0	0.019
10.3678570	*Cc10t04730.1* C2H2-type domain-containing protein	0	0.018
8.30746352	*Cc08t16130.1* Nucleoside diphosphate kinase	0	0.017
10.3612160	*Cc10t04630.1* BHLH domain-containing protein	0	0.014
**Intermediate, yield of green beans—Bootstrap Forest**			
4.234906	*Cc04t00320.1* Conserved hypothetical protein	0	0.019
10.1467983	*Cc10t01960.1* Regulator of chromosome condensation (RCC1) family protein	0	0.012
11.1400701	*Cc11t00520.1* Exopolygalacturonase	+2297	0.01
11.30547303	*Cc11t13750.1* Galectin domain-containing protein	−6340	0.01
11.20731087	*Cc11t05390.1* *NB-ARC* domain-containing protein	0	0.0092

SNPs were ranked based on their variable importance (“Portion”) in the models, with the corresponding genomic locations shown in Figure 4. The table lists the SNP identifier (SNP ID); the nearest gene and its putative function; the distance in base pairs between the SNP and the gene’s start site (0 indicates the SNP is within the gene and negative and positive distances correspond to upstream and downstream positions relative to the TSS); and the importance score from the model.

## Data Availability

Publicly available datasets were analyzed in this study. These data can be found here: Ferrão et al. [13] (https://doi.org/10.5061/dryad.1139fm7). Additional processed data are contained within the article and Appendix A.

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
