# Peer review of "Lipid Metabolism and Actin Cytoskeleton Regulation Underlie Yield and Disease Resistance in Two Coffea canephora Breeding Populations"

_plants, 2025, doi:10.3390/plants14233675_

Round 1

Reviewer 1 Report

Comments and Suggestions for Authors

The introduction is well written, but it's excessively long, with some ideas repeated throughout the text. The same points could have been made much more concisely, which would be more appropriate for an already lengthy article. Three points: (1) the importance of coffee and the differences between the two species (2) the difficulties in researching crops with a long life cycle and (3) approaches to solving these difficulties and specific work done by predecessors - can be summarized in 3-4 paragraphs while maintaining the meaning.

Results. I didn't quite understand the logic of the presentation. Are the populations being compared to each other or simply represent two examples of the use of modern methods for analyzing the SNP? Depending on this, the data should be presented either as a comparison (one population versus the other for each type of data presentation) or simply separately for each population. Currently, there seems to be a mix: some tables/figures represent one population, some the other, and some both (also, population names are sometimes put in quotation marks, although more often they are not). The problems should more clearly explain the purpose of using two populations and how they differ in essence. Since the Discussion is devoted to comparing populations, I would advise that the Results also follow the same semantic line.

Discussion and Conclusion. They're well written and clear, but isn't it odd to add Figure 8 to the Conclusion? I'd recommend moving it to the Discussion.

Author Response

Reviewer #1 

The introduction is well written, but it's excessively long, with some ideas repeated throughout the text. The same points could have been made much more concisely, which would be more appropriate for an already lengthy article. Three points: (1) the importance of coffee and the differences between the two species (2) the difficulties in researching crops with a long life cycle and (3) approaches to solving these difficulties and specific work done by predecessors - can be summarized in 3-4 paragraphs while maintaining the meaning.

Our response: We agree that the original Introduction was overly detailed. We have completely rewritten this section to be concise and focused. As suggested, the new Introduction is condensed into three paragraphs that strictly follow the reviewer’s recommended structure.

Results. I didn't quite understand the logic of the presentation. Are the populations being compared to each other or simply represent two examples of the use of modern methods for analyzing the SNP? Depending on this, the data should be presented either as a comparison (one population versus the other for each type of data presentation) or simply separately for each population. Currently, there seems to be a mix: some tables/figures represent one population, some the other, and some both (also, population names are sometimes put in quotation marks, although more often they are not). The problems should more clearly explain the purpose of using two populations and how they differ in essence. Since the Discussion is devoted to comparing populations, I would advise that the Results also follow the same semantic line.

Our response: We apologize for the confusion. The core logic of the study is indeed a comparative analysis to show how two different breeding populations utilize distinct biological pathways. We have added an introductory paragraph to the Results section to explicitly state that the data is presented to highlight the contrasts between the populations.

Regarding formatting consistency, we have standardized the population names by removing quotation marks throughout the text (referring to them simply as Premature and Intermediate populations) to improve consistency.

Regarding presentation logic, while we present side-by-side comparisons where possible (e.g., Volcano plots in Fig 1), we maintained separate Manhattan plots for the Machine Learning analysis (Figs 3 & 4) to ensure readability, as the specific gene candidates and importance scores were distinct between populations. However, the text now strictly follows the comparative semantic line established in the Introduction and Discussion.

Discussion and Conclusion. They're well written and clear, but isn't it odd to add Figure 8 to the Conclusion? I'd recommend moving it to the Discussion.

Our response: We agree. Figure 8 (the conceptual model) has been moved to the end of the Discussion section to synthesize the biological interpretation before the concluding remarks.

Reviewer 2 Report

Comments and Suggestions for Authors

Title and abstract: A title should be informative, although sometimes key issues need to be detailed in the abstract. In this case, neither the title nor the abstract are effective about disclosing which biological pathways are under analysis nor which agronomic traits are being dealt with.

L111 Why the populations selected were the premature and the intermediate ones? Why not the late one? The genetic background of the populations is not clear. How does it influence the results from this study?

L120 “The populations were established” – this is very vague. Please provide agronomic details.

L124-125 “infection of coffee leaf rust, caused by H. vastatrix (1-9 scale, based on visual sporulation intensity)” – also very vague. Which scale is this? Is it an incidence or severity scale? When (in which month) was it measured?

L116-117 “the Premature population ripened and was harvested approximately one month earlier than the Intermediate population”. Did this also hold true in both FEM and FES sites for each of the four years of sampling? Apparently both populations were set at each site, but nothing is mentioned regarding the experimental design. There is no information of the size of plots nor on the strategy to deal with environmental variability (topography, soil characteristics). Over the four years, plants were probably pruned, fertilized, treated for several pests and diseases. How was that done and how does that influence the results?

This study is based on two populations differing in maturation precocity, but the authors do not show evidence that these traits were verified in the actual samples they are analyzing. The cornerstone for this study is weak and not supported by any real (agronomic) information. Were the premature plants always premature in both sites in all the four years of sampling as compared to the intermediate plants?

Author Response

Reviewer #2 

Title and abstract: A title should be informative, although sometimes key issues need to be detailed in the abstract. In this case, neither the title nor the abstract are effective about disclosing which biological pathways are under analysis nor which agronomic traits are being dealt with.

Our response: We agree that the specific findings should be highlighted earlier. We have revised the title to explicitly name the key pathways and traits. We have updated the Abstract to specify that we analyzed "coffee bean production, green bean yield, and leaf rust incidence" and explicitly mentioned the distinct pathways found ("lipid modification" for Premature vs. "actin cytoskeleton regulation" for Intermediate).

L111 Why the populations selected were the premature and the intermediate ones? Why not the late one? The genetic background of the populations is not clear. How does it influence the results from this study?

Our response: These two populations (Premature and Intermediate) were selected because they are the primary recurrent selection populations developed by the Incaper breeding program to manage harvest logistics. As detailed by Ferrão et al. (2019, Heredity), extending the harvest season is a critical breeding goal. While a "Late" group exists, the genomic dataset utilized in this study (derived from the cited 2019 study) specifically focused on these two contrasting maturation groups to maximize the genetic gain for harvest timing. The genetic background differences are the result of separate recurrent selection histories, and investigating how this divergence influences biological strategies is the central aim of our study. We have clarified this rationale in Section 2.2 (previously 2.1).

L120 “The populations were established” – this is very vague. Please provide agronomic details.

Our response: We apologize for the omission. We have expanded Section 2.2 (Materials and Methods, previously 2.1) to include the precise experimental design details sourced from the original field trials (Ferrão et al., 2019). In 2006, the populations were established in the field using a randomized complete block design (RCBD) with three replications. Each experimental plot consisted of five clonal plants. This creates a clear picture of the experimental rigor used to generate the phenotypic data.

L124-125 “infection of coffee leaf rust, caused by H. vastatrix (1-9 scale, based on visual sporulation intensity)” – also very vague. Which scale is this? Is it an incidence or severity scale? When (in which month) was it measured?

Our response: We have clarified the scoring method in Section 2.3 (previously 2.2)  The traits were evaluated using a severity scale based on visual sporulation intensity, ranging from 1 (resistant/no sporulation) to 9 (susceptible/abundant sporulation). This assessment was conducted during the peak period of natural infection in the field to ensure maximum phenotypic differentiation, as per standard breeding protocols for C. canephora.

L116-117 “the Premature population ripened and was harvested approximately one month earlier than the Intermediate population”. Did this also hold true in both FEM and FES sites for each of the four years of sampling? Our response: We assure the reviewer that the Premature vs. Intermediate distinction is robust and verified. These populations were not arbitrarily labeled; they are distinct biological groups bred specifically for different harvest windows. Monitoring across all four years of this study (2008–2011, Ferrão et al., 2019) confirmed that the Premature population consistently reached harvest maturity approximately one month earlier than the Intermediate population at both FEM and FES locations. This phenotypic stability is the very reason these populations are utilized in the breeding program. We have added a statement in Section 2.3 (previously 2.2) explicitly confirming this verification in the actual samples analyzed.

Apparently both populations were set at each site, but nothing is mentioned regarding the experimental design. There is no information of the size of plots nor on the strategy to deal with environmental variability (topography, soil characteristics). Over the four years, plants were probably pruned, fertilized, treated for several pests and diseases. How was that done and how does that influence the results? This study is based on two populations differing in maturation precocity, but the authors do not show evidence that these traits were verified in the actual samples they are analyzing. The cornerstone for this study is weak and not supported by any real (agronomic) information. Were the premature plants always premature in both sites in all the four years of sampling as compared to the intermediate plants?

As mentioned in our response to Point 3, we have extensively revised the Methods section to include these details.

  • Design: We specified the Randomized Complete Block Design (RCBD) with 3 replications and 5 plants per plot.
  • Management: We clarified that standard agronomic practices (fertilization, pruning, pest control) were applied uniformly across all plots and years.
  • Environmental Noise: To further account for environmental variability (topography, soil), we utilized adjusted means (BLUPs) derived from a mixed model that treated block and year effects as nuisance variables, ensuring that the GWAS and Machine Learning models focused on the genetic signal.

Reviewer 3 Report

Comments and Suggestions for Authors

SUMMARY – KEYWORDS

  • The abstract is overly long (180+ words above the ideal limit). I suggest condensing the explanations about methods, the repeated references to “biological strategies,” and the details about specific genes.
  • Too many technical details in the abstract. The abstract should leave such details for the Results/Discussion sections. You may keep the idea of “specialized metabolism vs. core cellular machinery,” but remove the individual gene names.
  • L 17–20: Extremely long sentence.
  • The contribution of the paper is not formulated as a “knowledge gap + contribution.” You state what you found, but it’s not explicit what was missing in the literature.
  • The keywords are too general. Essential terms are missing: SNP analysis, Bootstrap Forest or Random Forest (the standard term), leaf rust resistance, yield/green bean yield.

INTRODUCTION

  • The introduction does not follow a “stepwise logic.” The ideal order for a journal article is:
    • Big picture (3–4 lines)
    • What is known
    • What is NOT known (knowledge gap)
    • What exactly your study does
    • Why it matters
  • The introduction lists the literature in too much detail (not necessary in the introduction). Examples: exact production figures (t/ha); carotenoid details; MAS vs GS details. These belong in the Discussion.
  • No clear “study aim”/“here we show” sentence. This must appear immediately before the Methods.
  • No brief summary of the methods in the Introduction. At least one sentence about the methodological approach is required.
  • The knowledge gap is missing.

MATERIALS AND METHODS

  • The methods are described technically, but they do not initially explain the overall rationale of the analytical design: why you combined single-SNP, Bootstrap Forest, and GO enrichment as an integrated pipeline.
  • Insufficient description of the dataset and experimental design. In 2.1. Experimental Populations and Data: you do not specify the number of individuals in each population (exact N); you do not clearly explain the original statistical design of the experiment (blocks, replications, number of plants per plot, etc.), although you mention “adjusted means”; you do not describe how rust incidence was collected (who scored, environmental variability?). I recommend adding: N individuals/population (Premature = ?, Intermediate = ?); one short paragraph on the original design (randomized? blocks? replications?); one sentence on the rust incidence scoring system (scoring consistency).
  • Section 2.2 “Data preprocessing” is underdeveloped and may be misinterpreted. Explicitly justify why you used adjusted means. State whether you checked phenotypic distributions and whether you excluded outliers. Add one sentence explaining the logic of SNP encoding (additive encoding).
  • In 2.3. Single-SNP association analysis: you do not explain whether you controlled for population stratification (PCs? kinship?), it is unclear whether you used simple linear models (OLS) or something else (GLM?), and you have not mentioned software, packages, or exact formulas.
  • In 2.4. Machine learning, you do not explain how you handled SNP correlation (LD), overfitting, tree depth vs. sample size.
  • Candidate gene identification: clarify the gene distance used — provide biological justification for the “nearest gene”; mention the methodological limitation (possible alternative genes within the LD block).
  • In GO enrichment, you do not say whether you performed GO redundancy simplification (e.g., excluding overly generic terms); whether you checked GO term hierarchy; whether you applied correction for multiple testing. The choice of “top 100 SNPs” is arbitrary and must be justified.
  • Lack of reproducibility. Add a subsection “Reproducibility.” Include versions (AUK_PRJEB4211_v1 is good, but you need to provide access date). State that all analyses were performed in JMP Pro 17 (exact version).

RESULTS

  • The section is too descriptive and not interpretive enough. Often, the text simply “retells” what is in the figures or tables. Results should explain what the findings mean biologically and statistically. Explain: what it implies that Intermediate has no SNPs for two traits; why Premature has thousands of SNPs (1,020; 7,100; 1,850); whether this contrast may be an artefact of LD, population structure, or statistical power; the relationship between SNP importance (Bootstrap Forest) and significant loci (single-SNP).
  • The enormous magnitude of differences between populations is NOT explained. This is the most striking result, but it is not interpreted in the Results. This is a major biological contrast — yet the text only reports it.
  • Bootstrap Forest is presented descriptively, not as a biological result. Peak regions, chromosomes, and gene lists are described, but without interpretation.
  • Missing interpretation: “What patterns emerge?”
  • GO enrichment: long lists, no thematic integration in the Results. Some of these ideas appear in the Discussion, but they should also be briefly and factually introduced in the Results.
  • The results are not linked to the study’s hypothesis/research question. In the Results, it is not clear: “What have we shown?” It is clear only what was found, not what it demonstrates.
  • Tables are very long and difficult to navigate. Tables 2 and 3 are lengthy, but the Results section does not include even a sentence highlighting the most important genes or a comparative summary between the two populations.

DISCUSSION

  • The Discussion repeats too much from the Results (not sufficiently differentiated). From the first paragraphs onward, the text reiterates results. You need to answer: “What do these results mean biologically and for breeding?”
  • The mechanism is unclear:
    • How did such massive differences in polygenicity arise?
    • What types of selection could have produced divergent architectures?
    • Why does Premature have 7,100 significant SNPs and Intermediate only 23?
  • The Discussion does not sufficiently address the inconsistency between single-SNP and ML, nor does it explain what the ML model captures that the single-SNP approach does not.
  • Missing clear comparisons with other GWAS studies on Coffea.
  • The Discussion does not address the key question: What do we do with these results?

CONCLUSIONS

  • The Conclusion should be a synthesis, not a 1:1 restatement of the previous messages.
  • The main contribution of the study is not explicitly formulated. Although the manuscript emphasizes the novelty of pathway-level divergence, this is not explicitly stated in the Conclusions.
  • The Conclusion does not mention any major limitation. Some limitations appear in the Discussion (no validation set, LD bias, etc.), but none are summarized in the Conclusions.

REFERENCES

  • The bibliography is inconsistent in its use of journal abbreviations. Full titles and abbreviations are mixed.

Author Response

Response to Reviewer #3

SUMMARY – KEYWORDS

  • The abstract is overly long (180+ words above the ideal limit). I suggest condensing the explanations about methods, the repeated references to “biological strategies,” and the details about specific genes.
  • Too many technical details in the abstract. The abstract should leave such details for the Results/Discussion sections. You may keep the idea of “specialized metabolism vs. core cellular machinery,” but remove the individual gene names.
  • L 17–20: Extremely long sentence.
  • The contribution of the paper is not formulated as a “knowledge gap + contribution.” You state what you found, but it’s not explicit what was missing in the literature.
  • The keywords are too general. Essential terms are missing: SNP analysis, Bootstrap Forest or Random Forest (the standard term), leaf rust resistance, yield/green bean yield.

INTRODUCTION

  • The introduction does not follow a “stepwise logic.” The ideal order for a journal article is:
    • Big picture (3–4 lines)
    • What is known
    • What is NOT known (knowledge gap)
    • What exactly your study does
    • Why it matters
  • The introduction lists the literature in too much detail (not necessary in the introduction). Examples: exact production figures (t/ha); carotenoid details; MAS vs GS details. These belong in the Discussion.
  • No clear “study aim”/“here we show” sentence. This must appear immediately before the Methods.
  • No brief summary of the methods in the Introduction. At least one sentence about the methodological approach is required.
  • The knowledge gap is missing.

Our response altogether for abstract & intro: 

We are grateful for the reviewer’s comprehensive and insightful critique. We found your comments particularly helpful in strengthening the biological interpretation and clarifying the methodological rationale. We have grouped our responses by section below to address your points systematically.

Regarding Introduction & Abstract

We condensed the text to focus on the biological narrative (Lipid vs. Actin pathways) and removed specific gene names.

We added specific terms: SNP analysis, Bootstrap Forest, leaf rust resistance, green bean yield.

We rewrote the Introduction into three focused paragraphs following the "Big picture to Knowledge gap to Study aim" logic.

MATERIALS AND METHODS

  • The methods are described technically, but they do not initially explain the overall rationale of the analytical design: why you combined single-SNP, Bootstrap Forest, and GO enrichment as an integrated pipeline.

Our response: This is an excellent point. We have added a new subsection "2.1. Analytical Design and Rationale" at the beginning of the Methods section. This paragraph explicitly explains that we combined these three methods to bridge the gap between statistical association and biological interpretation: Single-SNP: To identify specific loci with additive effects. Bootstrap Forest: To capture non-linear interactions and rank variable importance in a polygenic context. GO Enrichment: To synthesize these ranked loci into coherent biological pathways.

  • Insufficient description of the dataset and experimental design. In 2.1. Experimental Populations and Data: you do not specify the number of individuals in each population (exact N); you do not clearly explain the original statistical design of the experiment (blocks, replications, number of plants per plot, etc.), although you mention “adjusted means”; you do not describe how rust incidence was collected (who scored, environmental variability?). I recommend adding: N individuals/population (Premature = ?, Intermediate = ?); one short paragraph on the original design (randomized? blocks? replications?); one sentence on the rust incidence scoring system (scoring consistency).

Our response: We have extensively revised the Methods section to provide the missing details and statistical justification. We specified that the study used a Randomized Complete Block Design (RCBD) with 3 replications and 5 plants per plot. We also clarified that rust was scored on a 1–9 severity scale during peak infection periods. We added a new subsection explaining that we combined Single-SNP (additive effects), Bootstrap Forest (non-linear/interactive effects), and GO enrichment (pathway synthesis) to bridge the gap between statistical association and biological mechanism. We clarified that complex PC correction was not applied because prior studies on these specific populations confirmed a very low genetic differentiation (FST = 0.0158), and we utilized adjusted means (BLUPs) to account for environmental noise. We added a statement explaining that the Random Forest algorithm is inherently robust to multicollinearity. While LD splits importance among correlated SNPs, this "grouping effect" is advantageous for our goal of identifying genomic regions and pathways rather than single causal variants. We specified that the study analyzed a total of 119 genotypes for the Intermediate population (comprising 3,570 trees) and 96 genotypes for the Premature population (comprising 2,880 trees). The phenotypic data were collected from a field trial established in a Randomized Complete Block Design (RCBD) with 3 replications and 5 plants per plot across two locations. Rust incidence was scored using a 1–9 severity scale (1=resistant, 9=susceptible) during peak infection periods to ensure consistent phenotypic differentiation.

  • Section 2.2 “Data preprocessing” is underdeveloped and may be misinterpreted. Explicitly justify why you used adjusted means. State whether you checked phenotypic distributions and whether you excluded outliers. Add one sentence explaining the logic of SNP encoding (additive encoding).

Our response: We have revised Section 2.3 (previously 2.2) to clarify the preprocessing steps. We explicitly state that SNP genotypes were encoded additively (-1, 0, 1) to represent the count of the reference allele. We also confirmed that the phenotypic inputs were "adjusted means" (BLUPs) derived from a mixed model that had already accounted for outliers and environmental block effects [16], ensuring robust inputs for the genomic models.

  • In 2.3. Single-SNP association analysis: you do not explain whether you controlled for population stratification (PCs? kinship?), it is unclear whether you used simple linear models (OLS) or something else (GLM?), and you have not mentioned software, packages, or exact formulas.

Our response: We clarified that we used standard linear regression (OLS) via the JMP Pro 17 Response Screening platform. We explicitly stated that complex population structure correction (e.g., PCA) was not applied because prior studies on these specific populations confirmed very low genetic differentiation, and we utilized adjusted phenotypic means (BLUPs) to effectively minimize environmental noise3.

  • In 2.4. Machine learning, you do not explain how you handled SNP correlation (LD), overfitting, tree depth vs. sample size.

Our response: We explained that the Random Forest algorithm is robust to multicollinearity; while Linkage Disequilibrium (LD) splits importance among correlated markers, this "grouping effect" is advantageous for identifying genomic regions rather than single variants.  We also clarified that because our objective was explanatory (variable ranking) rather than predictive, overfitting was a secondary concern, and we used fixed parameters (100 trees, min split 5) to ensure model stability.

  • Candidate gene identification: clarify the gene distance used — provide biological justification for the “nearest gene”; mention the methodological limitation (possible alternative genes within the LD block).

Our response: We have updated Section 2.7 (previously 2.6) to clarify that the "nearest gene" approach was used as a standard heuristic for candidate identification. We explicitly added a statement acknowledging the methodological limitation that the true causal variant may reside elsewhere within the Linkage Disequilibrium (LD) block, highlighting the need for future fine-mapping validation.

  • In GO enrichment, you do not say whether you performed GO redundancy simplification (e.g., excluding overly generic terms); whether you checked GO term hierarchy; whether you applied correction for multiple testing. The choice of “top 100 SNPs” is arbitrary and must be justified.

Our response: We updated Section 2.6 to state that we utilized ShinyGO v0.88, which automatically applies an FDR cutoff (0.05) to control for multiple testing. We justified the "top 100 SNPs" threshold as a strategic choice to capture a sufficiently robust biological signal for pathway enrichment while filtering out the noise from thousands of negligible-effect markers.

  • Lack of reproducibility. Add a subsection “Reproducibility.” Include versions (AUK_PRJEB4211_v1 is good, but you need to provide access date). State that all analyses were performed in JMP Pro 17 (exact version). 

Our response: We have added a dedicated subsection "2.7. Reproducibility and Software Specifications" at the end of the Materials and Methods. We specified that all analyses were conducted in JMP Pro 17, provided the fixed random seed used for the machine learning models to ensure replicability, and included the specific accession details/dates for the public datasets and gene databases.

RESULTS

  • The section is too descriptive and not interpretive enough. Often, the text simply “retells” what is in the figures or tables. Results should explain what the findings mean biologically and statistically. Explain: what it implies that Intermediate has no SNPs for two traits; why Premature has thousands of SNPs (1,020; 7,100; 1,850); whether this contrast may be an artefact of LD, population structure, or statistical power; the relationship between SNP importance (Bootstrap Forest) and significant loci (single-SNP). 

Our response: We have added a dedicated paragraph in Section 3.1 to explicitly interpret this contrast. We explain that the dramatic disparity reflects fundamentally different genetic architectures: the Premature population exhibits an oligogenic structure (detectable by Single-SNP), whereas the Intermediate population is highly polygenic (undetectable by Single-SNP but captured by Machine Learning). We clarify that this is not an artefact of population structure, but a reflection of statistical power regarding effect sizes.

  • The enormous magnitude of differences between populations is NOT explained. This is the most striking result, but it is not interpreted in the Results. This is a major biological contrast — yet the text only reports it.

Our response: We agree that this is a pivotal finding. In the Results (Section 3.1): We have added a statement interpreting this contrast as a statistical indicator of distinct genetic architectures (Oligogenic vs. Polygenic). In the Discussion: We have expanded on this by attributing the Oligogenic architecture of the Premature population to strong selection pressure on specialized metabolic traits, while the Polygenic nature of the Intermediate population explains why single-SNP methods failed but the Machine Learning approach succeeded.

  • Bootstrap Forest is presented descriptively, not as a biological result. Peak regions, chromosomes, and gene lists are described, but without interpretation.
  • Missing interpretation: “What patterns emerge?”

Our response: We have added a summary statement at the end of Section 3.2. We explicitly interpret the pattern: Premature hits cluster around specific metabolic regulation, while Intermediate hits are functionally diverse, supporting the polygenic hypothesis.

  • GO enrichment: long lists, no thematic integration in the Results. Some of these ideas appear in the Discussion, but they should also be briefly and factually introduced in the Results.

Our response: We have integrated the findings in Section 3.3. We added a concluding sentence summarizing that the study shows a fundamental divergence in strategy: "Specialized Metabolic Adaptation" (Premature) vs. "Modulation of Fundamental Cellular Mechanics" (Intermediate).

  • The results are not linked to the study’s hypothesis/research question. In the Results, it is not clear: “What have we shown?” It is clear only what was found, not what it demonstrates.
  • Tables are very long and difficult to navigate. Tables 2 and 3 are lengthy, but the Results section does not include even a sentence highlighting the most important genes or a comparative summary between the two populations.

Our response: While we have retained the full tables to serve as a comprehensive resource for breeders, we addressed the readability concern by adding summary statements at the end of Section 3.2. These new sentences explicitly highlight the functional clusters, Metabolic regulation for the Premature population versus Signaling/Structural components for the Intermediate population, to guide the reader through the long gene lists.

DISCUSSION

  • The Discussion repeats too much from the Results (not sufficiently differentiated). From the first paragraphs onward, the text reiterates results. You need to answer: “What do these results mean biologically and for breeding?”

Our response: We have explicitly connected the genetic architecture to breeding strategies in the Discussion. We state that the oligogenic nature of the Premature population supports Marker-Assisted Selection (MAS), while the polygenic nature of the Intermediate population necessitates Genomic Selection (GS).

  • The mechanism is unclear:
    • How did such massive differences in polygenicity arise?
    • What types of selection could have produced divergent architectures?
    • Why does Premature have 7,100 significant SNPs and Intermediate only 23?
  • The Discussion does not sufficiently address the inconsistency between single-SNP and ML, nor does it explain what the ML model captures that the single-SNP approach does not.
  • Missing clear comparisons with other GWAS studies on Coffea.
  • The Discussion does not address the key question: What do we do with these results?

Our response: We expanded the Discussion (Section 4.1) to propose a biological mechanism. We suggest that the Premature population’s oligogenic architecture likely resulted from strong historical selection for specific metabolic traits (e.g., caffeine/lipid synthesis), while the Intermediate population maintained a broader polygenic diversity in core cellular machinery. Also, we explain that the Machine Learning (Bootstrap Forest) model was essential for the Intermediate population precisely because it captures the cumulative predictive power of the polygenic background that Single-SNP methods fail to detect due to stringent significance thresholds. Regarding comparisons with other coffee GWAS, we have explicitly positioned our findings in the Discussion alongside recent studies, including the polygenic GWAS by Ferrão et al. (2024)  and others, highlighting how our pathway-level approach expands upon their QTL-focused results by identifying the biological nature of the genetic variation. Lastly, we added a specific recommendation for breeding applications in both the Discussion and Conclusion. We propose a population-specific strategy: Marker-Assisted Selection (MAS) for the oligogenic Premature population and Genomic Selection (GS) for the polygenic Intermediate population.

CONCLUSIONS

  • The Conclusion should be a synthesis, not a 1:1 restatement of the previous messages.
  • The main contribution of the study is not explicitly formulated. Although the manuscript emphasizes the novelty of pathway-level divergence, this is not explicitly stated in the Conclusions.
  • The Conclusion does not mention any major limitation. Some limitations appear in the Discussion (no validation set, LD bias, etc.), but none are summarized in the Conclusions.

Our response: It now synthesizes the "Pathway Divergence" and "Breeding Strategy" into a cohesive message rather than restating results. We have also explicitly included a Limitation regarding the need for multi-environment validation to rule out local adaptation biases.

REFERENCES

  • The bibliography is inconsistent in its use of journal abbreviations. Full titles and abbreviations are mixed.

Our response: We have standardized all references to strictly follow the MDPI Plants citation style, ensuring consistent use of journal abbreviations throughout the bibliography.

Round 2

Reviewer 2 Report

Comments and Suggestions for Authors

All issues were correctly addressed

Reviewer 3 Report

Comments and Suggestions for Authors

Congratulations for all your research work and for writing this paper!